# Crystal Structure Complexity and Approximate Limits of Possible Crystal Structures Based on Symmetry-Normalized Volumes

**DOI:** 10.3390/ma17112618

**Published:** 2024-05-29

**Authors:** Oliver Tschauner, Marko Bermanec

**Affiliations:** 1Department of Geoscience, University of Nevada, Las Vegas, NV 89154, USA; 2Institute of Geological Sciences, University of Bern, 3012 Bern, Switzerland; marko.bermanec@gmail.com

**Keywords:** crystal structure prediction, volume prediction, crystal symmetry index, cage structures, battery materials

## Abstract

Rules that control the arrangement of chemical species within crystalline arrays of different symmetry and structural complexity are of fundamental importance in geoscience, material science, physics, and chemistry. Here, the volume of crystal phases is normalized by their ionic volume and an algebraic index that is based on their space-group and crystal site symmetries. In correlation with the number of chemical formula units Z, the normalized volumes exhibit upper and lower limits of possible structures. A bottleneck of narrowing limits occurs for Z around 80 to 100, but the field of allowed crystalline configurations widens above 100 due to a change in the slope of the lower limit. For small Z, the highest count of structures is closer to the upper limit, but at large Z, most materials assume structures close to the lower limit. In particular, for large Z, the normalized volume provides rather narrow constraints for the prediction of novel crystalline phases. In addition, an index of higher and lower complexity of crystalline phases is derived from the normalized volume and tested against key criteria.

## 1. Introduction

The relation between crystal structure and composition of chemical species is at the heart of the science of condensed matter, whether it is the chemistry or physics of solids, the mineralogy and petrology of rocks, or material engineering. The problem may be cast in the principal question of whether there is a general relation that correlates symmetry, volume, and composition of all chemically possible compounds and their solid solutions. More specifically, it is asked for actual limits for possible crystal structures and the existence of forbidden zones in the correlation of symmetry, density, and composition. The existence of such limits is evident from the existence of the gaseous and the liquid state, but it is worth exploring if, within a range of plausible densities, narrower constraints are obtainable. This question receives additional interest through the search for new large, porous, multi-component structures, which are essential in the chemical industry as catalysts or catalyst matrices [1,2], detergents [3], filter materials [3,4], battery membrane materials [5], to mention just a number of applications.

More recently, powerful algorithms have been developed that provide crystal structures for a given chemical compound in a given unit cell shape [6] and, thus, address part of the question that we ask here, although under the constraint of fully occupied lattice sites, given cell shape, and for pure compounds only. Computational assessment of structures with very large unit cells, such as those of nano- and mesoporous materials, is computationally costly.

Within the range of crystalline materials, minerals pose additional problems but also serve as a repository for material synthesis: There is barely a material of industrial importance that does not occur in nature or has natural analogs. As they occur in nature, minerals are commonly multi-component phases that have formed within multiphase systems [7,8]. Commonly, minerals carry minor and trace element concentrations. While applied solid-state science has focused on synthesis with limited sets of chemical ingredients for a long time, the wealth of multivariant solid solutions in natural systems provides a vast repository of materials that guide material science where properties are controlled by dopants.

The distribution of crystalline phases among the seven crystal systems has been assessed with statistical measures [9,10,11,12]. Hummer [12] observed that within uncertainty, the distribution of mineral species among the 32 point groups obeys a power law. The assessment of structural complexity has been based on the distribution of atoms on distinct sites [13], computational weight [14], or, quite successfully, based on network topologies and probability [15,16]. The problems of the statistics of symmetry distribution and of the complexity of structures are related but not equivalent: A statistics of symmetry that is based on the seven crystal systems neglect essential features of crystal complexity. For instance, phases of structures as simple as gold and as complex as the zeolite paulingite are thus ranked as equally highly symmetric because both are cubic. Statistics of the symmetries of crystalline phases that is based on point groups [12] is more distinctive, but it still places phases like sphalerite, ZnS, and zunyite, Al_13_Si_5_O_20_(OH,F)_18_Cl, into one category because they assume the same point group, although their structural complexity is vastly different in terms of the sizes of their asymmetric units.

The distribution of crystalline species across space groups and compositional ranges and the size of their asymmetric units implicitly carry information about symmetry-based constraints on possible structures, but none of the measures of symmetry and complexity that are mentioned above make these constraints explicit. The network topology of structures [15,16] provides quantitative rankings of structural complexity, but by principle, it does not carry information about the energetic constraints on possible structures, which are, foremost, controlled by their density at given conditions of pressure and temperature. Consequently, this approach does not provide predictions on possible structures and volumes for a given chemical composition, which is the question that we ask in this paper.

Here, it is shown that a combination of unit cell volume, ionic volume, and an algebraic index of crystal complexity provide tentative upper and lower limits of possible crystalline phases as a function of Z, the number of chemical formula units. In addition, an index of the complexity of crystal structures can be derived from this correlation that does not reduce actual crystal structures but is based on Wyckoff multiplicities, Z, and ionic volume

## 2. Materials and Methods

First, a general reference volume of chemical species that disregards sterical or symmetrical constraints and possible asymmetry of bond polyhedra is defined by the total crystal ionic volume ([17], hereafter: ‘ionic volume’). The crystal radii represent the radial part of the electron wavefunctions of bonded atoms [18] and are based on a vast set of empirical data for each ion [19]. Thus, the cubes of the crystal radii are the primary space-filling entity for any given structure, while the angular part of the wavefunctions and geometric constraints cause deviations from closed packings. Thus, the reference ionic volume abstracts from the geometric constraints and the directional dependence of valence electron distributions, and it is defined here for a compound A_i_B_j_C_k_… where i, j, k, … gives the stoichiometry of the chemical species A, B, C, … as
(1)$$V_{ion}=4\pi /3(i\cdot r_{A}^{3}+j\cdot r_{B}^{3}+k\cdot r_{C}^{3}+\cdots )$$
and the radii r_A,B,C_,… are the crystal radii for given valence and coordination and spin state [19]. Although this reference ionic volume neglects directional contributions to bonds and any sterical and geometrical constraints that avail for crystalline phases, it correlates with the unit cell volume V_uc_ and the number of chemical formula units Z as
(2)$$V_{uc}=1.87\left(7\right)\cdot Z\cdot V_{ion}$$
with an adjusted R^2^ of 0.989 for all minerals and phases that are listed in Table 1. The correlation is shown in Figure 1.

In this comparison, minerals and phases with unit cell volumes between 70 and 9 × 10^4^ Å^3^ and Z between 1 and 192 were selected, including sulfides, arsenides, oxides, silicates, borates, phosphates, and arsenates (Table 1). It is noteworthy that the correlation is quite good considering the wide variety of composition, structure, and size of the asymmetric units of the minerals and phases examined here (Table 1). On the other hand, the mean deviation of ~39% is significant. This large mean deviation is intrinsic to the large variation in structures and composition of the materials that are compared here, whose specific structural differences find an indirect expression in the discrepancies between ionic and actual volume. The concept of correspondence of states in the sense of the van der Waals equation finds its limitation for solids here. This fact, but in combination with the observation of an overall strong correlation between ionic and unit cell volumes, is used here as a basis for a more general assessment of structural complexity.

In the second step, an algebraic index of the intrinsic symmetry of the phases has to be defined. The criteria for a useful index are chosen as follows:(1)The index should correlate with the complexity of structures within structure families, e.g., the coupled substitution that derives bixbyite- and pyrochlore-type phases from fluorite-type oxides or tetrahedrite-type phases from the sphalerite-structure should be reflected by the index.(2)Polymorphs that are the result of structural transitions that obey the Landau criteria should also have higher indices. For instance, the transitions from cubic to rhombohedral and from cubic to tetragonal to orthorhombic ABO_3_–perovskites should be reflected by the index.


The above two criteria provide straightforward measures of the merit of a crystal symmetry index because they are based on well-defined concepts of the structural evolution of solids. In addition, it is required that:
(3)The index should generally scale with increasing structural complexity as defined by the size of the asymmetric unit, and, more specifically, it should rank solids that assume the same space group but with vastly different sizes of their asymmetric units accordingly higher or lower.


Here, the following symmetry index is defined:(3)$$I_{SG}=\frac{\sum_{i}n_{i}\cdot M_{i}\cdot {SOF}_{i}}{M_{max}\cdot \sum_{i}n_{i}}$$
where n_i_ is the number of occupied sites for chemical species i, M_i_ is the Wyckoff multiplicity of that site, M_max_ is the maximal Wyckoff multiplicity that is possible in that space group, and SOF_i_ is the site fraction occupancy of the site by species i.

Thus, I_SG_ is the ratio of the sum of the orders of the subgroups of a space group that correspond to occupied sites in a structure, divided by the order of the space group itself times the number of occupied sites, that is, the lowest possible symmetry and maximal multiplicity. I_SG_ is an isomorphic mapping from the space groups onto the field of rational numbers. The numbers quantify the deviation of the assumed symmetry from the lowest possible symmetry. Thus, generally I_SG_ ≤ 1. Generally, the higher the complexity of a structure, the closer I_SG_ is to unity. Partially vacant sites reduce I_SG_. In particular, I_SG_ = 1 for any structure in space group 1 as long as all sites are fully occupied because all M_i_’s equal M_max_; this is independent of the size of the asymmetric unit and, thus, limits the information about structural complexity that is represented by I_SG_.

This point and the functionality of I_SG_ as a measure of structural complexity and discriminator for symmetry reductions, in general, is illustrated by some examples: The case of α- and β-quartz is illustrative. With Equation (3), I_SG_ is
(4)$$I_{SG}\;(\alpha )=\frac{3+6}{6+6}=\frac{3}{4}$$
for α-quartz and
(5)$$I_{SG}\;\left(\beta \right)=\frac{6+12\times 0.5}{12+12}=\frac{1}{2}$$
for β-quartz; therefore, the high-temperature, higher-symmetric polymorph is assigned to a smaller index and, thus, lesser structural complexity, reflecting the higher vibrational part of the free energy of β-quartz compared to α-quartz. It is noted that the lower index I_SG_ of β-quartz results from the partial occupancy of site 12c, which, however, is essential in establishing the higher symmetry of the beta-phase. Thus, the index I_SG_ behaves in accordance with the crystal physics of the two quartz polymorphs. For pyrite and marcasite, one finds I_SG_ = ¼ and ¾, respectively (Table 1), which is in agreement with the crystal physics of these two polymorphs of FeS_2_. Substituting half of S with As gives arsenopyrite, FeAsS, with reduced crystal symmetry and I_SG_ = 1 (Table 1). Further examples are given in Table 1: arsenopyrite has index 1, but so do enargite, panguite, anorthite, and, as mentioned, any crystalline phase that assumes space group P1 with all sites fully occupied.

Thus, I_SG_ quantifies higher or lower symmetry within groups of related structures but does not discriminate between the intrinsic symmetry of very different structures. This issue is addressed further below in the Discussion. However, first of all, the results show that in combination with relation (2), I_SG_ is instrumental in assessing limits for possible structures that are approximately quantitative (and it further outlines a path for making them fully quantitative).

## 3. Results

Correlating V, V_ion_, and I_SG_ with Z, the number of chemical formula units defines allowed and forbidden fields of possible crystal structures independent of composition (as far as explored here). This is shown in Figure 2: Relation (2) is recast as ratio V_uc_/V_ion_ but multiplied with the inverse of I_SG_ to give a normalized, dimensionless volume (hereafter: ‘symmetry-normalized volume’). In addition, the symmetry-normalized volume V_sym_ is weighted by the average slope of (2).
(6)$$V_{sym}\equiv \frac{1}{1.87\cdot I_{SG}}\cdot \frac{V_{uc}}{V_{ion}}$$

V_sym_ is related to Z, the number of chemical formula units per cell, as shown in Figure 2. As implied by the large deviations of individual values around the linear correlation (2), the values of V_sym_ occupy a large range of values between ≤2 and 1000. However, Figure 2a also shows rather well-defined upper and lower bounds for possible correlations between V_sym_ and Z. Figure 2b depicts the same data in a linear-logarithmic plot to better show the distribution of V_sym_ at small Z. Furthermore, the examined data (Table 1) indicate a distribution statistic for crystalline species within those bounds. The upper and lower bounds are discussed first.

**Figure 2 materials-17-02618-f002:**
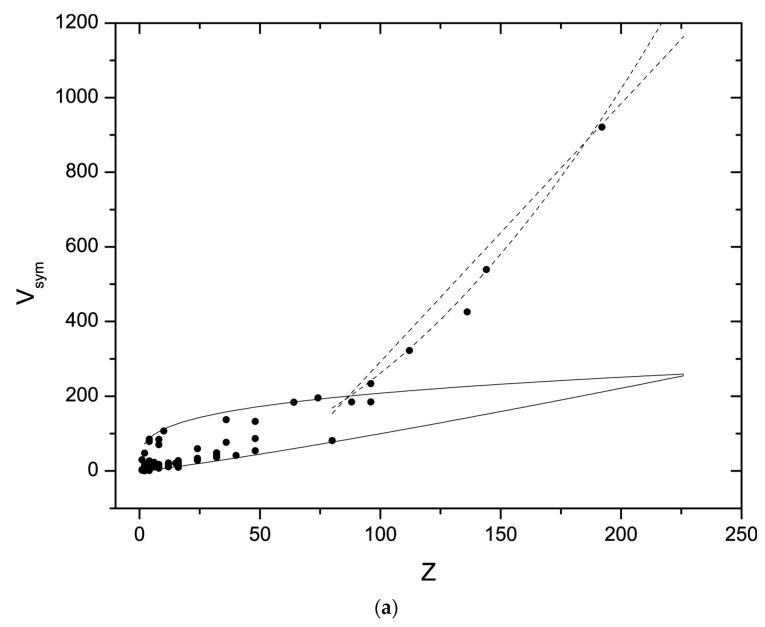
(**a**) Correlation of normalized volume V_sym_ = V_uc_/(1.87 I_SG_V_ion_) and the number of chemical formula units Z, based on the data given in Table 1. Extensive forbidden zones of mechanically unstable low-density structures and of overly compact high-density structures are visible. Straight lines represent the upper and lower boundaries between the allowed and the forbidden zones. These lines are calculated based on the equations given in Table 2. The dashed lines represent the tentative limits for the low-density limit at large Z > 80 (Table 2). (**b**) Same but in linear-logarithmic plotting to make the data at low V_sym_ and Z better visible. The meaning of the lines and symbols is the same as in (**a**).

**Table 2 materials-17-02618-t002:** Tentative upper and lower boundaries of the correlation between Z and symmetry-normalized volume. The upper bounds were obtained by fits through the uppermost data points in Figure 2. For the lower bound, a fitted equation was modified so that it does not cut through the smallest Z data. The lower bound may have more terms that are influential at small Z, but this cannot be assessed using the given data. The adjusted R^2^ of fits are 0.97, 0.99, and 0.92, respectively (top to bottom), and the linear relation has not been fitted.

Boundary	Z	V_sym_
Lower *	>1	0.5 Z^1.15^
Upper	1–80	60 Z^0.27^
	>80	0.03 Z^1.97^
		or: 6.92 Z−400

* There is probably another term that limits this boundary to the symmetry-normalized volume around 2.

## 4. Discussion

### 4.1. Constraints on Possible Crystal Structures, Statistics, and Crystal Structure Prediction

Figure 2 shows large apparently forbidden zones of volume–symmetry correlations for inorganic compounds. This is less surprising than it may seem: for instance, the value of V_sym_ = 10 at Z = 48 falls within the lower forbidden zone. For a hypothetical phase of silica, these values give a density of 16.5 g/cm^3^ if I_SG_ = 1, and for a cubic metric, for I_SG_ = 0.75 and Z = 24, the density would be 11 g/cm^3^ in a cubic cell. In the upper forbidden zone, with, for example, Z = 48 and I_SG_ = 1, a cubic phase of silica with V_sym_ = 300 assumes a density of 0.52 g/cm^3^. Thus, the forbidden zones represent regions where the available Wyckoff sites do not allow for stable structures either because at low density they are mechanically unstable toward collapse into denser structures or at high density they require extreme compression (very high inner energy, respectively), over which structures of smaller Z and lower density are favourable.

It is worth examining the boundaries between the allowed and the forbidden zones because they define the range where within inorganic crystalline structures are possible. Since the data set is limited (Table 1), these upper and lower boundaries are tentative. No strict evaluation of their functional dependence on Z and their uncertainties is attempted here. The approximate boundaries that are depicted in Figure 2 are summarized here in Table 2: All boundaries follow approximate power-law dependences of Z and provide tentative constraints on possible structures. In particular, at large Z, there are strong constraints on structures that are close to the high-density limit, and such crystalline phases may occur only sporadically. For instance, for a hypothetical polymorph of silica at Z = 144, V_sym_ = 146, and with I_SG_ = 0.5625, the volume of the unit cell is about 2300 Å^3^, and a phase in space group Nr. 228 would have a density around 6.5 g/cm^3^ with Si and O on sites of multiplicities 48 + 96, and 96 + 192, respectively. I_SG_ = 1 gives about twice the volume, but no cubic space group is compatible with this symmetry index, Z, and a stoichiometry 1:2. However, a rhombohedral phase of silica with Z = 144 and all atoms on special positions is compatible with the upper limit for a density of 3.5 g/cm^3^, which is between coesite and stishovite. These assessments shall only illustrate the rather strong constraints that are imposed by the high-density limit of the correlation Z ∝ V_sym_ (Figure 2, Table 2) and, hence, are made without any consideration of physical bond distances and -angles, which pose additional constraints, nor with any consideration of the free energy of such very large and dense polymorphs of silica.

The low-density limit is a positive monotonous correlation between the symmetry-normalized volume V_sym_ and Z, which simply states that the higher the number of chemical formula units, the larger the unit cell volume of possible crystalline structures that is stable towards spontaneous collapse into denser structures. The low-density limit obeys a power law with a slope of ~0.27 up to about Z = 80, where the slope changes to almost 2 (Table 2). However, instead of a fitted power-law relation for these data above Z = 80, a limiting linear relation (Table 2) is also consistent with the observed data (Figure 2). The divergence between the Z^1.97^ and the linear limiting relation becomes significant above Z = 200. This uncertainty about the actual functional relation between V_sym_ and Z above Z = 80 is a consequence of the limited statistics and emphasizes that the limiting relations in Table 2 are approximate. In any case, it is clear from Figure 2 that there is a change in slope for the low-density limit of the V_sym_-Z correlation.

The extrapolation of the small-Z 0.27-power limit intersects the high-density limit around Z = 225 (Figure 2), implying that no crystalline structures are possible beyond that value, if these limits would hold for any Z. However, the crossover from 0.27 to ~2 widens the field of possible structures between the upper and lower boundary for Z > 90. Moreover, Figure 2 shows that the convergence of the Z^0.27^ small-Z, low-density limit towards the high-density limit causes a narrowing of the field of possible structures before the crossover of the power of the Z-dependence.

### 4.2. Effect of Pressure and Temperature

It was shown recently [22] that the measured pressure-dependent volumes and the pressure-dependent ionic volumes are strictly linearly correlated (at least for oxides and at least above 1–3 GPa) because the non-linear compressibility beyond Hooke’s law is dominated by the non-linear compression of the anions [17]. Thus, the ratio V_uc_/V_ion_ remains constant upon compression within narrow uncertainties and does not shift any of the phases in Figure 2.

Since ionic volumes are temperature-invariant by definition, V_sym_ increases with increasing temperature. Because thermal expansivity remains generally small up to the 1 bar melting point, V_sym_ remains below the upper limit of Figure 2. Examples are corundum at 300 and 2170 K; forsterite, Mg_2_SiO_4_, at 300 and 948 K; and protoenstatite, MgSiO_3_, at 1633 K (Table 1). It is expected that crossing the upper limit of the Z-V_sym_ relation is correlated with structural rearrangements or with melting, but this remains to be examined for a larger set of materials and data obtained at sufficiently high temperatures.

### 4.3. Statistical Distribution of Structures

The examined data provide only limited statistics for the count of structures across Z and structural complexity. It appears that at Z between 1 and 20, the maximum of existing structures is close to the high-density limit of the symmetry-normalized volume (Figure 2), and this is physically quite plausible because the crystallizing compounds are expected to minimize volume under the given sterical constraints imposed by the composition, ion size, and directional distribution of valence electrons. At large Z, it appears that the few available hydrogen-free inorganic solids are closer to the low-density limit than the high-density limit—in fact, no material appears to be close to the high-density limit for Z > 100. Based on the examples that are given above, this is plausible: the upper limit implies rather high densities for most compounds and space groups, thus limiting the number of possible structures. Figure 3a depicts the distribution of different chemical compound classes within the allowed range of Z-V_sym_. Simple oxides and sulfides are more abundant in the low-Z range, where sulfides appear to be closer to the high-density oxides closer to the low-density limit. Silicates with small Z occupy the middle to close to the high-density limit but extend to large Z where they are close to—or at the low-density limit, whereas framework structures based on phosphate-, arsenate- or molybdate-groups are closer to the high-density limit up to Z ~ 80. In light of the discussion in Section 4.1., this observation provides a tentative explanation for, or at least a quantitative assessment of, the comparatively lesser number of such framework structures compared to those built from silica and alumosilicate networks.

The apparent high count of small Z silicates of higher density compared to simple oxides is not expected but finds a tentative explanation by the easier accommodation of constituent vacancies in high-temperature simple oxides, but this observation may well be biased by the very limited statistics.

The possible occurrence of very dense phases with very high Z is limited to accordingly very high pressures. Inside Earth, however, the geothermal gradient poses an entropic constraint on the formation of such phases. An additional constraint arises from the mobility of chemical elements in depth of Earth where no partial melts occur and where only metasomatosis through supercritical fluids acts as a potential means of mobilizing and segregating less common elements [23]. Moreover, chemically bonded water augments the number of potential mineral species in general [11,24].

**Figure 3 materials-17-02618-f003:**
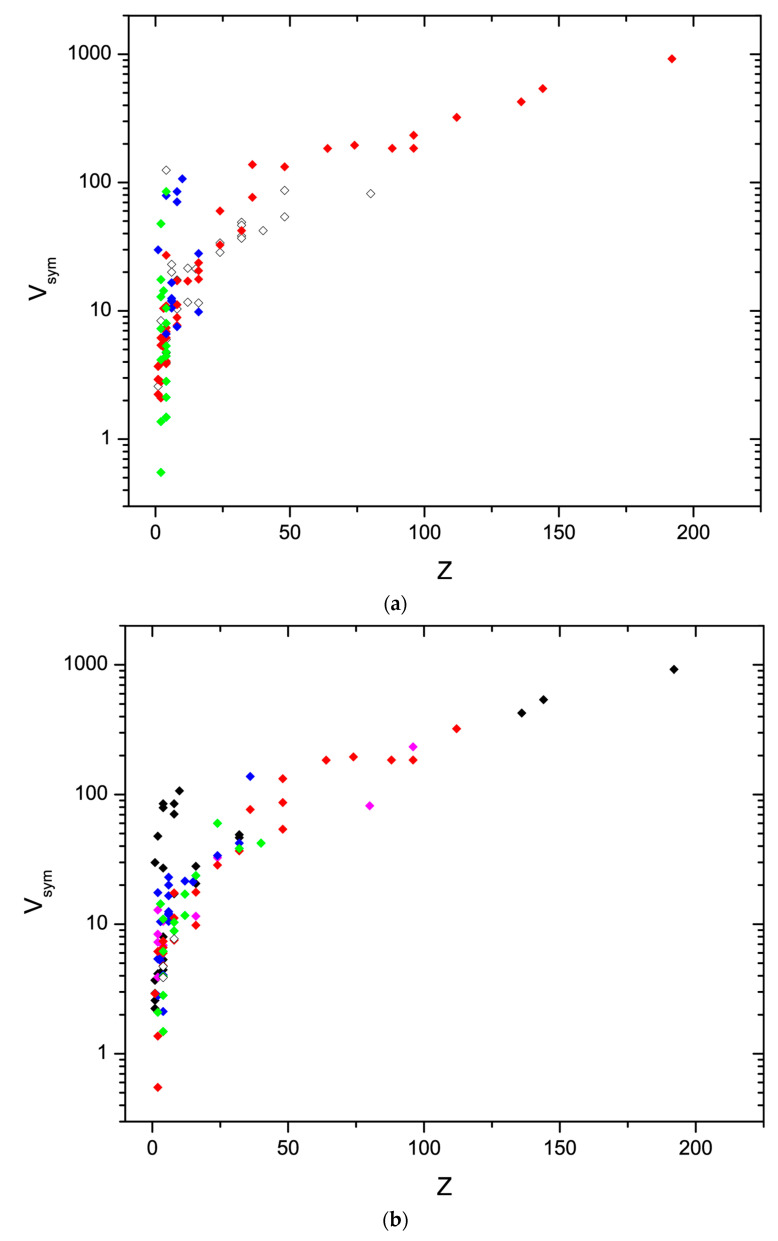
(**a**) Figure 2 with key for different compound groups: red = silica and silicates, blue = simple oxides, titanates, aluminates; green = sulfides and arsenides. All others: Hollow diamonds. The remaining high Z data that are not silicates are phosphate-, arsenate-, borate-, or molybdate framework structures. As indicated in the discussion, these phases fall closer to the high-density limit of the Z-V_sym_ correlation than the silica-based large frameworks. (**b**) Figure 2 with key for different crystal systems: black = cubic; magenta = tetragonal; blue = trigonal and hexagonal; red = orthorhombic; green = monoclinic; hollow diamonds = triclinic. No strong correlation is visible for the examined set of data. There appears to be a tentative preference for cubic symmetry among the high V_sym_ phases and a preference for monoclinic phases at small Z and close to the high-density limit (see Table 2). The small number of triclinic phases is likely a bias from the limited number of phases and the exclusion of compounds with H-bonds [24,25]. An apparent preference for high crystal symmetry at high Z is noteworthy and deserves future study.

### 4.4. Index of Crystal Complexity

It was stated in Methods that I_SG_ does not discriminate between structurally unrelated crystalline phases. It should also be noted that V_uc_/V_ion_ does not provide a measure of crystalline complexity along the lines defined in Methods. For instance, Sc_2_O_3_ in the cubic bixbyite-type structures has a higher ratio V_uc_/V_ion_ than panguite, Ti_2_O_3_, a ‘collapsed bixbyite’, that is clearly the more complex structure, according to the three criteria established in Methods. Similarly, the V_uc_/V_ion_ of rhombohedral ABO_3_ perovskite LaAlO_3_ is higher than that of the orthorhombic perovskite CaTiO_3_. Thus, V_uc_/V_ion_ does not match the fundamental criteria of a physically meaningful index, and this statement holds for a variety of combinations of parameters that suggest plausible indices at first glance.

However, a general measure of structural complexity is obtained by combining I_SG_ with the number of chemical formula units Z and the ionic volume as:(7)$$I_{SG}\cdot Z\cdot \frac{V_{ion}}{4\pi /3\cdot r_{B}^{3}}=I_{cmplx}$$
where r_B_ is the Bohr radius. Obviously, I_cmplx_ is closely linked to relation (2) and to the correlation between Z and the symmetry-normalized volumes in Figure 2. Based on an algebraic index, I_SG_, and being dimensionless, I_cmplx_ serves itself as an index. The term Σn_i_ in the denominator of I_SG_ is divisible by Z, and (6) may be reformulated accordingly, but for clarity, it is better to keep Z as an explicit parameter. The full list of I_cmplx_’s is given in Table 1.

With I_cmplx_ as a measure of structural complexity, the ambiguity of the unity values of the index I_SG_ for structures with all atoms on sites of lowest possible symmetry in a space group vanishes. This and the other properties of I_cmplx_ are illustrated through a number of examples that serve as tests of the three criteria established in Methods. 

The minerals, enstatite (MgSiO_3_), anorthite (CaAl_2_Si_2_O_8_), sapphirine, and the clathrate framework silica phase Si_264_O_528_ all assume I_SG_ = 1 despite obviously different structural complexity but their I_cmplx_’s are 403, 746, 724, and 39,889, respectively (Table 1), thus discriminating these structures in a quantitative fashion through assigning a single but unique number to each of them, in accordance with the third criterion in Methods. 

The polymorphic Ca-alumosilicates anorthite and dmisteinbergite have both I_SG_ = 1, but *I_complx_* assumes the values 1202.27 and 283.74 (Table 1), respectively, thus discriminating the ordered, low symmetric polymorph anorthite from the less ordered, high-temperature polymorph dmisteinbergite. Thus, I_cmplx_ matches the fine-scale ranking of complexity that is obtained through topological network analysis [15] in this and in many other cases. 

Corundum assumes I_cmplx_ = 111.62 but ilmenite I_cmplx_ = 115.48, representing the symmetry breaking imposed by the splitting of the cation site, in accordance with the first and second criterion of a meaningful index (see Section 2). γ-Al_2_O_3_ gives I_cmplx_ = 40.36, which correctly reflects the lower complexity of this high-T polymorph of Al_2_O_3_ compared to corundum. The Ga_2_O_3_-type κ-Al_2_O_3_ has I_cmplx_ = 331 due to an asymmetric unit that is large compared to corundum and to its low space group symmetry. A phase with low symmetry but modestly large asymmetric unit like the sodium chromate lopezite assumes I_SG_ = 1 because all atoms reside on special positions; its I_cmplx_ is 536.16, which ranks this Na-chromate between the inosilicates enstatite and clinoenstatite (both MgSiO_3_, see below and Table 1). This seems plausible since the structures of the latter two are based on tetrahedral chains with larger cations residing between these chains on distorted polyhedra with higher coordination, whereas in lopezite dimers of tetrahedra are arranged in linear arrays, like chains where every third tetrahedron is omitted, with likewise large low symmetric polyhedra occupied by Na. 

For the series of K-Al-silicates kalsilite, nepheline, and panunzite, the I_cmplx_’s are 64.09, 446.23, and 2030.3, respectively, in accordance with the different sizes of the asymmetric units. I_SG_ of marcasite, FeS_2_, and cubanite, CuFe_2_S_3_, are both 0.75, but their I_cmplx_’s are 102.8 and 324.1, correctly representing the difference in structural complexity between an AB_2_- and an ACB_2_-stoichiometry and a rutile-derived versus a sphalerite-derived tetrahedral superstructure. Isocubanite, (Cu,Fe)S, which is isotypic with sphalerite, ZnS, ranks very low with I_cmplx_ = 6.0 as a highly symmetric and disordered high-T phase of very low complexity. 

It is worth noting that I_cmplx_ ranks complexity not always according to expectations that are based on the crystal metric. For instance, high-temperature clinoenstatite has lower I_cmplx_ than enstatite despite its lower crystal symmetry. However, as a high-temperature phase, clinoenstatite has ~½ the unit cell of enstatite, which forms by condensation of phonons of the clinoenstatite lattice. Thus, I_cmplx_ reflects the thermodynamic relation between both polymorphs of inosilicate MgSiO_3_ correctly (or more specifically, the first and second Landau criterion, and this is owed to the factor Z in I_cmplx_). The same statement holds for perovskites and for order-disorder induced symmetry breaking, such as for CoAsS, as well as for symmetry-reducing coupled substitutions for thorianite- and sphalerite-derived structures (see Table 1). Hence, I_cmplx_ is an indirect measure of degrees of freedom in structures and fulfills criteria 1 and 2 that were outlined in Methods. Further examples of high–low T polymorphs and the according relations of their I_cmplx_’s are listed in Table 1.

### 4.5. Limitations of I_cmplx_

As defined, the index I_cmplx_ is limited to solids where crystal radii can be applied at least as limiting cases of bonding, and that excludes proper metals and molecular materials. It has been shown previously [26] that at least pressure-induced polymorphism in elemental metals obeys simple relations between their volumes, number of valence shell electrons, and their principal and orbital quantum numbers, quite similar to the relations found for the radial part of valence states in ions that define crystal radii [18,27,28]. Thus, it is very likely that the concept of I_cmplx_ can be expanded to metals and alloys.

Hydrogen poses a different problem. The apparent crystal radius of H^+^ is negative [19], which indirectly represents the effect of the H-bond averaged for many inorganic crystalline species. However, this effect should initially not be included in the ionic volumes as defined here. Therefore, in the present paper, the H-bearing phases are not considered. V_sym_ of H-bearing phases is expected to show a systematic shift toward lower values compared to non-H-bearing phases, which is indirectly expressed in the apparent negative crystal radius of H^+^ [19]. I_cmplx_ is expected to be shifted to larger values [25].

Finally, molecular materials are not considered here and are expected to deviate systematically by the role of dissipative forces that control the intermolecular distances. It is noted that apparent ionic volumes of metallorganic network phases show a similar relation to actual molar volumes upon compression as inorganic phases [22], despite the markedly directional bonding in those compounds. Thus, the calculation of an apparent ionic volume of molecular phases may still be instructive but is beyond the scope of this study.

### 4.6. Tentative Statistics of Crystalline Phases Based on I_cmplx_

Due to the rather limited set of crystallographic data examined here, the distribution of I_cmplx_ among the inorganic compound classes remains tentative: simple sulfides and oxides have overall lowest values of I_complx_ between 1 and 200 (Table 1), complex sulfides assume in part much higher values. Salts with non-polymerized complex anions like carbonates and chromates have low to modest complexity with I_complx_’s of 50 to 300 (Table 1). Silicates assume values mostly between 200 and 1000, with the exception of framework structures, which extend to numbers as high as 40,000 (Table 1, Figure 2). Similarly, high numbers are obtained by frameworks that are based on molybdate, phosphate, arsenate, and borate groups. The apparent greater richness of large frameworks of silicates and phosphates compared to arsenates and molybdates is tentatively explained through the fact that the high-density limit of possible structures poses rather narrow constraints at large Z, while the latter two substance classes are placed closer to this upper than the lower density limit. This distribution is depicted in Figure 3a, which shows the same data as Figure 2 but with a key for different chemical compound classes.

It is noted that this tentative classification by I_cmplx_ agrees well with that by Krivovichev [15]. Even the numerical values are mostly quite similar. This is not incidental because the network topology of crystalline phases is strongly correlated with the homotopic structure defined by the subgroup splitting of space groups that underlies the symmetry index I_SG_ and, thus, I_cmplx_. A more rigorous mathematical discussion of this point is beyond the frame of this paper.

No strong correlation is visible for the distribution of crystal systems across V_sym_ for the examined set of data. This is shown in Figure 3b. There appears to be a tentative preference for cubic symmetry among the high V_sym_ phases and a preference for monoclinic phases at small Z and close to the high-density limit (see Table 2). The small number of triclinic phases is likely a bias from the limited number of phases and the exclusion of compounds with H-bonds [24,25]. An apparent preference for high crystal symmetry at high Z is noteworthy and deserves future study.

## 5. Conclusions

A general correlation between unit cell and ionic volume and an algebraic index I_SG_ of crystal structure space group symmetry are combined into a symmetry-normalized volume V_sym_ that defines approximate upper and lower limits for possible crystal structures. These limits are tentatively fitted as power laws of Z. The lower limit is defined by the mechanical stability of low-density structures and follows a power law that crosses over from power 0.27 to either a linear or a nearly quadratic relation above Z = 80. In the relationship between Z and V_sym_, most of the examined materials crystallize in small Z structures fall close to the upper limit, whereas it is shown that at large Z, the upper limit is more prohibitive for the formation of structures of physically possible densities and structures close to the low-density limit appear favourable. Based on these observations, an index of crystal structure complexity I_cmplx_ is defined that properly scales with parameters such as order-disorder processes, symmetry reduction by coupled substitutions, 2nd order transitions, topological hierarchy, and number of atoms in the asymmetric unit. I_cmplx_ is based on Wyckoff multiplicities, Z, and ionic volume only. It allows for categorizing classes of compounds by average complexity.

## Figures and Tables

**Figure 1 materials-17-02618-f001:**
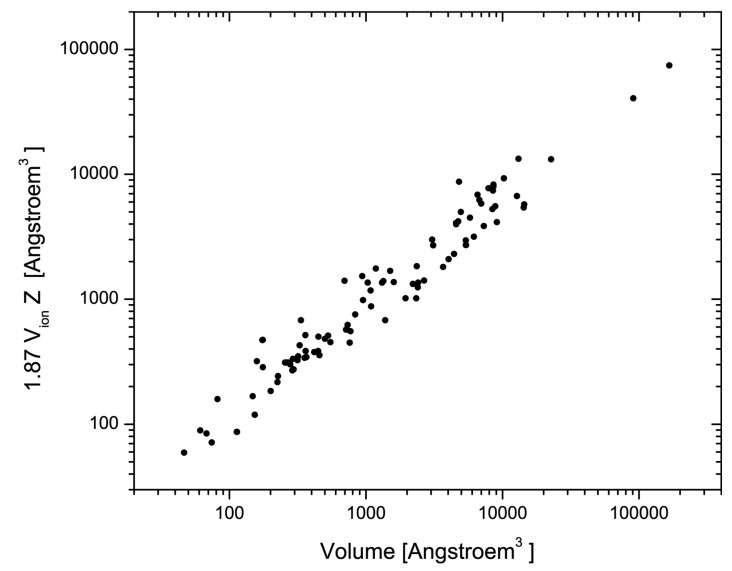
Correlation between ionic and unit cell volume, both in Å^3^. The correlation is based on the inorganic solids and minerals given in Table 1.

**Table 1 materials-17-02618-t001:** List of examined phases, Z, symmetry index I_SG_, measure of structural complexity I_cmplx_, ionic, and unit cell volumes. The records are ordered by increasing complexity index I_cmplx_ (see Section 4). For natural crystalline phases, the mineral names are given; for zeolite frameworks, the common name is given; I_SG_ and I_cmplx_ are calculated according to Formulas (3) and (6). Ionic volumes are calculated based on Equation (1) using crystal radii from [19]. Unit cell volumes are averages based on data given in [20]. Because of ubiquitous polymorphism and in agreement with common practice, mineral names are used for phases that occur as such. Information about formation conditions is found in ref. [21].

Phase	Z	I_cmplx_	I_SG_	V_ion_ [Å^3^]	V [Å^3^]
CaSiO_3_ Davemaoite	1	2.13872	0.03472	38.23544	74.1
Cu_1−x_Fe_x_S_4_ Isocubanite	4	5.99751	0.04167	22.3364	148.04
FeS, NiAs-type	2	6.01746	0.07833	23.84129	61.19
ThO_2_ Thorianite	4	7.66796	0.03125	38.0769	175.92
FeS Mackinawite	2	9.08926	0.125	22.56732	67.92
SiO_2_ Stishovite	2	9.58896	0.1875	15.872	46.61
SiO_2_ β-Quartz	3	28.0573	0.5	15.48049	113.53
(Ni,Fe)_9_S_8_ Pentlandite	2	37.25551	0.03186	362.9161	1032.6
γ-Al_2_O_3_	10	40.36082	0.09722	25.7689	500.94
MgAl_2_O_4_ Spinel	8	42.76356	0.09722	34.12799	527.28
SiO_2_ α-Quartz	3	56.1146	0.75	15.48049	113.25
KAlSiO_4_ Kalsilite	2	64.0879	0.40404	49.22811	200.51
(Fe,Ni)_3+x_S_4_ Smythite	3	64.91754	0.14583	92.10339	359.76
FeS_2_ Pyrite	4	68.52611	0.25	42.53511	159.04
MgCO_3_ Magnesite	6	72.06815	0.27778	26.8402	279.43
Cu_2_FeSnS_4_ Stannite	2	75.30975	0.25	93.49162	317.98
SiO_2_ α-Cristobalite	4	76.71165	0.75	15.872	153.15
CaCO_3_ Calcite	6	82.2767	0.27778	30.64215	366.63
CuFeS_2_ Chalkopyrite	4	95.73091	0.3333	44.57059	291.57
LaAlO_3_	6	102.0812	0.27778	38.01764	326.93
FeS_2_ Markasite	2	102.7892	0.75	42.53511	81.63
(Ca_1.29_, U_0.50_…) (Ti_1.09_, Nb_0.79_…) O_6_ (O_0.98_, F_0.02_) Betafite	8	105.448	0.10417	78.54114	1081.21
Al_2_O_3_ Corundum	6	111.6213	0.41667	27.71396	255.89
Al_2_O_3_ Corundum 2170K	6	111.6214	0.41667	27.71396	269.66
Al_2_O_3_ Corundum 1173	6	111.6222	0.41667	27.71396	260.6
CoAsS Pa3	4	112.4699	0.27777	62.83221	173.93
CaTiO_3_ Perovskite	4	117.0592	0.625	29.0641	224.63
CaCO_3_ Aragonite	4	130.7108	0.625	32.45361	226.97
CoAsS P23	4	134.9541	0.3333	62.83221	173.93
Mg_2_SiO_4_ Forsterite	4	135.4082	0.58333	36.02152	289.58
Forsterite 948K	4	138.2444	0.58333	36.77582	295.83
FeTiO_3_ Ilmenite	6	155.4843	0.55556	28.95314	315.84
Na_6_(AlSi O_4_)_6_ Sodalite, anhydrous	1	177.354	0.4583	240.2053	759.05
Mg_2_Al_3_(AlSi_5_O_18_) Cordierite	2	215.8368	0.4524	148.0691	771.68
SiO_2_ Moganite	12	276.1619	0.9	15.872	455.99
MgSiO_3_ Protoenstatite 1633K	8	276.573	0.83333	25.75101	447.57
Ca(Al_2_Si_2_O_8_) Dmisteinbergite	4	283.7405	0.48485	90.81244	333.67
Ca(Al_2_Si_2_O_8_) Svyatoslavite	2	292.6064	1	90.81244	355.23
SiO_2_ Coesite	16	319.9136	0.8214	15.10948	549.47
CuFe_2_S_3_ Cubanite	4	324.1224	0.75	67.06243	447.97
MgSiO_3_ Clinoenstatite (LP)	8	324.7162	1	25.19448	418.16
κ-Al_2_O_3_	8	331.0387	1	25.68504	361.3
Al_13_Si_5_O_20_(OH,F)_18_Cl Zunyite	4	339.0025	0.2788	188.6869	2665.61
K(AlSi_3_O_8_) Orthoclase	4	399.0778	0.8125	76.21943	715.15
CoAsS Cobaltite	4	404.9029	1	62.83221	173.93
FeAsS Arsenopyrite	4	407.2117	1	63.19049	175.46
Cu_6_(Cu_4_Fe_2_)Sb_4_S_13_Tetrahedrite	2	445	0.22917	602.65	1071.23
Na_3_K(Al_4_Si_4_O_16_)Nepheline	2	446.2287	0.90909	152.3392	726.25
Sc_2_O_3_ Kangite	16	470.4255	0.55556	32.8497	954.1
Si_24_O_48_	24	477.9892	0.81818	15.10948	1386.3
Na_4_Al_3_Si_9_O_24_Cl Marialite	2	482.812	0.6406	233.9121	1093.33
SiO_2_ Linde-L framework	1	513.061	0.45076	706.5085	2204.67
SiO_2_ Chabasite	36	525.7893	0.6	15.10948	2337.86
K_2_Cr_2_O_7_ Lopezite	4	536.1572	1	83.20006	733.82
Mg_3_Al_2_Si_3_O_12_ Pyrope	8	604.4317	0.41667	112.5537	1503.48
MgSiO_3_ Enstatite	16	649.4323	1	25.19448	832.49
NaMg_3_Al_6_(Si_6_O_18_)(BO_3_)_3_(OH)_3_(OH) Dravite	3	744.5194	0.631	244.1276	1600
SiO_2_ ZSM48	48	751.1308	0.64286	15.10948	2409.07
SiO_2_ PTS_24	4	755.4774	0.70536	166.2043	2395.23
SiO_2_ Ferrierite	36	788.6839	0.9	15.10948	1952.36
Cu_3_AsS_4_ Enargite	2	921.18	1	285.89	293.97
K_0.04_Al_0.06_Si_0.44_O STA-30	144	1018.365	0.5141	8.53856	4428.3
SiO_2_	64	1099.686	0.70588	15.10948	3680.06
Fe_7_S_8_	4	1132.578	0.9375	187.4685	696.84
Mg_4_(Mg_3_Al_9_)O_4_[Si_3_Al_9_O_36_] Sapphirine	4	1166.724	1	181.0504	1312.11
CaAl_2_Si_2_O_8_ Anorthite	8	1202.271	1	93.28329	1342
SiO_2_ SSZ_58	74	1315.5	0.7303	15.10948	4035.5
Ti_2_O_3_ Panguite	16	1317.65	1	51.11772	938.69
Fe_1−x_ S Pyrrhotite-5C	4	1441.106	0.951	235.1511	1181.93
KAlSi_2_O_6_ Leucite	16	1578.023	1	61.21882	2356
SiO_2_ EU1-framework	112	1846.86	0.67742	15.10948	6169.52
SiO_2_ ZSM11-framework	96	1911.957	0.81818	15.10948	5400.4
Na_2_B_2_Si_2_O_6_	24	1938.914	0.75	66.86172	3058
SiO_2_ Dodecasile	136	2009.82	0.6071	15.10948	7303.64
(K,Na)AlSiO_4_ Panunzite	32	2030.274	0.875	45.00784	3110.17
SiO_2_ ZSM5-framework 70K	88	2212.032	0.94738	16.46933	5394.38
SiO_2_ ZSM5-framework	96	2413.125	0.94738	16.46933	5386.53
Si_0.73_Al_0.27_O_2_ Linde-Y	192	2595.843	0.526	15.95449	14451.9
Cs_7_AsMo_8_O_30_	8	2677.822	0.5	415.5403	6773.13
Si_152_O_292_	1	2691.434	0.75676	2207.592	9088.23
K_3_YB_6_O_12_	15	2794.109	0.797	145.0726	4583.29
LiRbP_2_O_6_	32	3437.44	1	66.67709	4586.48
KCoPO_4_	48	3620.86	1	46.8233	4736.48
Pb_4_Zn_8_P_8_O_32_	8	3862.882	1	299.7181	5793.63
CuP_2_O_4_F_4_	48	4027.021	0.8889	58.5843	8454.3
SiO_2_ Faujasite	1	4206.314	0.9	2901.02	14292
Mn_11_Na_4_._35_(PO_4_)_9_	12	4254.594	0.891	246.9964	8859.67
Na_96_(BePO_4_)_96_	1	4258.52	0.74	3572.058	12,760.41
LiPbB_9_O_15_	24	4269.365	0.852	129.5997	6978.43
Cs_12_Zn_4_B_20_O_40_	4	4305.496	1	668.1203	4947.65
CsTi_2_P_3_O_12_	32	4681.997	0.7037	129.058	7891.3
CsTi_2_As_3_O_12_	32	4838.574	0.7037	133.3741	8608.82
Rb_3_Sc_2_As_3_O_12_	16	5756.523	0.765	291.9244	4805.33

## Data Availability

All data are listed in the tables in the paper.

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
