# Peer review of "Crystal Structure Complexity and Approximate Limits of Possible Crystal Structures Based on Symmetry-Normalized Volumes"

_materials, 2024, doi:10.3390/ma17112618_

Round 1

Reviewer 1 Report

Comments and Suggestions for Authors

This is an interesting paper, proposing a semi-systematic approach to understanding crystals symmetry in relation with the structure of constituent molecular species.

As far as I can understand, the authors discuss the correlation between a corrected volume-related ratio (Vsym, eq. 6) - defined with respect to an empirically defined symmetry index ISG - with the number of chemical formula units of a given crystal lattice. Figure 2 of the paper is the main result, exhibiting area of allowed or not allowed structures, thereafter commented by the authors.

Not being an expert of this field, I can only comment on the quality of presentation, readability of the text, significance of the methods and results to a generic audience. I find the work significant, but not an easy reading. In particular

- I think the the introduction should be enriched with a more detailed exposition of some of the alternative methods cited by the authors, with a clearer explanations of why and how, in their opinion, these approaches are lacking

- perhaps the logarithmic representation of Fig. 2, relegated to the SI, could be helpful in the main text.

In the end, I think that the contents and language of the papers make it of interest mainly for a restricted audience of geologists/mineralogists, but publication after minor revisions should be allowed.

Author Response

Reviewer 1:

Comment: This is an interesting paper, proposing a semi-systematic approach to understanding crystals symmetry in relation with the structure of constituent molecular species.

As far as I can understand, the authors discuss the correlation between a corrected volume-related ratio (Vsym, eq. 6) - defined with respect to an empirically defined symmetry index ISG - with the number of chemical formula units of a given crystal lattice. Figure 2 of the paper is the main result, exhibiting area of allowed or not allowed structures, thereafter commented by the authors.

Not being an expert of this field, I can only comment on the quality of presentation, readability of the text, significance of the methods and results to a generic audience. I find the work significant, but not an easy reading. In particular

- I think the the introduction should be enriched with a more detailed exposition of some of the alternative methods cited by the authors, with a clearer explanations of why and how, in their opinion, these approaches are lacking

Response: Thank you, we follow your and the editor's suggestion to augment the introduction for better clarification and by pointing to possible applications of the present work in material science. Additional references are given. We add a statement why the network topological approach does not answer to our question of defining constraints in large structure prediction.

Comment:  perhaps the logarithmic representation of Fig. 2, relegated to the SI, could be helpful in the main text.

Response: We agree - we place this figure now in the main text.

Comment: In the end, I think that the contents and language of the papers make it of interest mainly for a restricted audience of geologists/mineralogists, but publication after minor revisions should be allowed.

Response: We added more text to the introduction that, hopefully, clarifies the relevance for solid state synthesis in general. We stress that minerals are a repository for material science (Name one crystalline applied material that has not a natural analogue or exists as mineral!).

For better clarity we add the chemical formulas of the minerals that are mentioned in the text and indicate which are polymorphs.

Reviewer 2 Report

Comments and Suggestions for Authors

The submitted manuscript is entitled “Crystal structure complexity and approximate limits of possible crystal structures based on symmetry-normalized volumes”. The study is sound and the presented results are interesting for the materials science community.

Here is the list of minor issues that should be considered in the manuscript revision:

1) The authors should go through the manuscript for typos, e.g. line 9: “Geoscience” in uppercase, line 104, etc.

2) Line 95: The limitation of  ‘Correspondence of states’ for solids should be more deeply discussed for clarity.

3) Figure 2: What is the difference between dashed and solid lines?

Author Response

Reviewer 2:

Comment: The submitted manuscript is entitled “Crystal structure complexity and approximate limits of possible crystal structures based on symmetry-normalized volumes”. The study is sound and the presented results are interesting for the materials science community.

Here is the list of minor issues that should be considered in the manuscript revision:

Comment 1) The authors should go through the manuscript for typos, e.g. line 9: “Geoscience” in uppercase, line 104, etc.

Response: Thank you! Changed! We checked the manuscript for further typos.

Comment 2) Line 95: The limitation of  ‘Correspondence of states’ for solids should be more deeply discussed for clarity.

Response: Thank you, this is an important point. We added to the wording of this section for better clarification.

Comment 3) Figure 2: What is the difference between dashed and solid lines?

Response: Thank you for noticing this! We added an explanation to the Figure caption.

Reviewer 3 Report

Comments and Suggestions for Authors

1/ Z definition is not clear. Can you include an simple example?

2/In Table2: Can you include the quality parameters of the established correlations? Ihn this context, choose the best fit for z>80

3/ In line 225: Correct the expression Z1.97

4/ Inside Fig 2 or as note: Indicate which equations correspond to the proposed adjustments.

Comments on the Quality of English Language

It is OK

Author Response

Reviewer 3:

Comment 1:  Z definition is not clear. Can you include an simple example?

Response:  Z is the common symbol for the number of chemical formula units per unit cell. However, Z is also a common symbol for the nuclear charge number.

For better clarification, we add the term 'number of chemical formula units' to the figure captions and once in each chapter.

Nuclear charge numbers are not used explicitly here, whence no confusion should arise.

Comment 2: In Table2: Can you include the quality parameters of the established correlations? Ihn this context, choose the best fit for z>80

Response:  Thank you. We add this information. We keep the linear relation for reference.

Comment 3: In line 225: Correct the expression Z1.97

Response: Thank you - corrected!

Comment 4: Inside Fig 2 or as note: Indicate which equations correspond to the proposed adjustments.

Thank you - added!